# Cross-Context Lexical Analysis

## Abstract

We propose a general framework for performing cross-context lexical analysis; that is, analyzing similarities and differences in term meaning and representation with respect to different, potentially overlapping partitions of a text collection.

We apply our framework to three different tasks: semantic change detection (discovering words whose meanings changed over time), comparative lexical analysis over context (finding context-sensitive and context-*in*sensitive terms), and word representation comparison (investigating randomness inherent in word embeddings).

## 1 Introduction

Natural language is almost always used in a particular context (e.g., a particular time, location, or purpose), and thus the interpretation of a sentence, phrase, or word inherently depends on this context. Indeed, the whole subject area of pragmatics studies the ways in which context contributes meaning[1]. In this paper, we are interested in analyzing the variations of term meaning in different–but comparable–contexts and propose a general framework for performing cross-context lexical analysis (CCLA). We use CCLA to generally refer to any analysis of term meaning or term representation in different contexts, especially for understanding the differences and similarities in *multiple* contexts.

Due to the generality of the notion of context, CCLA can be useful in many ways. For example, when context is defined as the time period a piece of text is written, CCLA allows us to compare the meaning of a word in different periods and reveal how a word may have evolved over time (Hamilton et al., 2016). If context is defined as location, it would allow us to study variations in the meaning of a word over different locations, potentially revealing influences of some locations on others (Kulkarni et al., 2016).

In general, we can use *any* associated attribute values of text data—including metadata—as context to form a partition. For example, the institution of a research article's author can be used as a "context variable" to partition the articles based on institutions or regions in the world of their authors.

Any meaningful partitioning of text data may also be regarded as implicitly defining a context value for each partition; sentiment analysis may allow us to define a sentiment context so positive and negative sentences would be regarded as belonging to different categories.

We can characterize any term in a specific context by its similarity to other terms in corresponding contexts. The similarity can be computed in many ways, including (e.g.) with word embeddings. This gives us a context-specific "term similarity profile" for every term. These profiles for the same term computed from different contexts can be compared to analyze the variations of term meaning across contexts.

Traditionally, such cross-contextual analysis has been done on a "topic-level" basis (Zhai et al., 2004; Mei and Zhai, 2006). However, this is limiting because only word co-occurrence data can be used to estimate the model. Thus, including distributional similarity metrics (or any other representation) is not built-in, and it is not obvious how to include it in a probabilistic model in an easily-interchangeable way. Lastly, relying solely on word co-occurrence statistics (which are often unigrams) misses opportunities to examine context windows of adjacent terms, which could be

---

[1] https://en.wikipedia.org/wiki/Pragmatics

useful for capturing word sense or ambiguity.

CCLA can be used to perform analysis in three distinct ways:

(1) a **term focused** approach, where the emphasis is placed on mining the terms themselves with respect to the collection of contexts. For example, we could detect words whose meanings have shifted over time (which we explore in section 3), or compare dialects of the same language across different regions;

(2) a **score focused** approach, where the emphasis is placed on defining a scoring function over terms that can detect context-sensitive (representative) or context-insensitive (shared) terms (which we explore in section 4). This can be useful as a component in downstream tasks such as feature selection, transfer learning, and information retrieval; and

(3) an **annotation focused** approach, where the emphasis is placed on understanding how the annotations for words change as a function of the context used to derive the annotation. We explore this in section 5, where we analyze the stability of two well-known word embedding methods.

These focuses often intermix and overlap.

This paper is organized in the following manner. Section 2 formalizes cross-context lexical analysis. Sections 3-5 investigate concrete applications of CCLA and illustrate each of the three focuses described above. Section 6 shows related work and section 7 concludes the paper.

All source code from this work is made publicly available online[2]. All datasets used in our experiments are also freely and publicly available.

## 2 A Framework for Cross-Context Lexical Analysis

We now formally define the framework for cross-context lexical analysis. Critical to CCLA is the idea of a context view. We define a context view as a tuple $C_i = (V_i, f_i)$, where $V_i$ is a set of unique terms $w \in V_i$ and $f_i : V_i \to \mathcal{A}$ is an annotation function that maps words from the vocabulary set $V_i$ to some shared analysis space $\mathcal{A}$. Potential annotations could be term probabilities ($\mathcal{A} = [0, 1]$) or word vectors ($\mathcal{A} = \mathbb{R}^d$), depending on the eventual goal. Different contexts $C_i$ and $C_j$ may share word tokens, but each word's annotation is specific to its context. That is, the term $w = amazing$

[2]url redacted for anonymous submission

may occur in both $V_i$ and $V_j$, but $f_i(w)$ is not necessarily equal to $f_j(w)$. This allows us to compare the usage of the token $amazing$ respective to each context. We refer to the set of all contexts as $\mathbf{C}$.

The vocabularies for each context come from a backing set of text documents $D$. This may be a corpus in the conventional notion—like the IMDB movie reviews (Maas et al., 2011)—or it may be a collection of such corpora. Due to this flexible nature of context views, it is not a requirement that all contexts partition $D$; contexts may even overlap. Take the sentiment analysis dataset collection as an example: imagine that $D$ contains documents from both IMDB and Yelp[3]. If we set $\mathcal{A} = \mathbb{R}^d$, we could define the contexts that comprise $\mathbf{C}$ over $D$ in the following way: let $C_{YELP} = (V_{YELP}, f_{YELP})$, where $V_{YELP}$ is all the terms that occur in the Yelp business reviews and $f_{YELP}(w)$ yields a $d$-dimensional word vector for $w$ learned on the Yelp dataset; similarly let $C_{IMDB}$ have $V_{IMDB}$ as all of the terms that occur in IMDB and $f_{IMDB}(w)$ yield a $d$-dimensional word vector for $w$ learned on IMDB; $C_{POS}$ and $C_{NEG}$ can be defined similarly, with vocabularies and word vectors coming from only the positive and negative reviews across both datasets, respectively. We could add a background context $C_{ALL}$ with a vocabulary consisting of all terms used across both datasets and with word vectors learned on the union of both datasets.

The comparison operator $\Phi$ takes multiple contexts and outputs a list of (word, score) tuples for each term in the shared vocabulary:

$$\Phi(C_j, \ldots, C_k)$$
$$= \left\langle (w, \phi(w, C_j, \ldots, C_k) \mid w \in \bigcap_{i=j}^{k} V_i \right\rangle$$

where the scoring function $\phi$ is user-defined and task-specific. For example, if our task is to identify words used similarly across contexts, our scoring function can be specified to give high scores to terms whose usage is similar across the contexts.

The scored terms returned from $\Phi$ are able to be processed by operators such as `head` (return the highest-scored terms), `tail` (return the lowest-scored terms), and `average` (return the average scores of all the terms).

[3]https://www.yelp.com/dataset_challenge

As an example application, we can use disjoint temporal segments as our context views in a term-focused task. Let $C_1$ be the initial time period context and let $C_2$ be the final time period context. We wish to discover $w \in V_1 \cap V_2$ that underwent semantic change. We define a $\phi$ such that a given term with similar annotations across $C_1$ and $C_2$ will have a *higher* score, and a given term with different annotations across $C_1$ and $C_2$ will have a *lower* score. Thus, when we run head($\Phi(C_1, C_2)$) the result is the terms that changed the least; tail($\Phi(C_1, C_2)$) will show the terms that changed the most, i.e., underwent semantic change. We discuss this particular application scenario in more depth in the next section.

## 3 Analysis of Semantic Change

The evolution of word usage is a well-studied area in linguistics. Also known as *semantic change* or *diachronic analysis*, it has received attention in the NLP community, most recently by Kim et al. (2014), Kulkarni et al. (2015), and Hamilton et al. (2016). All three methods are based on word embedding similarity, and learn separate embeddings for distinct time periods. For a brief outline of each method, see section 6. With these techniques, we can discover how words such as *awful* change meaning over time. In the 1850s, it meant *solemn* or *majestic*, whereas in the 1900s it meant *terrible* or *horrible* (Hamilton et al., 2016). Detecting and analyzing these semantic shifts allows us to learn about the culture and evolution of language.

We next formalize the problem in the CCLA framework and compare our findings to previous results.

### 3.1 CCLA Formulation

In this task, we will use disjoint temporal segments as our context views in a term-focused task. Let $C_1$ be the initial time period context and let $C_2$ be the final time period context. We wish to discover $w \in V_1 \cap V_2$ that underwent semantic change.

We define the following scoring function:

$$\phi(w, C_1, C_2) = \cos(NN(w, C_1), NN(w, C_2))$$

where $NN$ finds the top-$k$ nearest neighbors of $w$ in $C_i$ (and their corresponding similarities) by using its $d$-dimensional word vector annotation $f_i(w) \in \mathbb{R}^d$. Since the word vectors are normalized to unit length, the nearest neighbors are calculated using a dot product.

Essentially, $\phi$ measures how similar the usage of a particular $w$ is across the two contexts. Thus, to find words whose usage changed the most (i.e., underwent semantic change), we find the $w$'s with the least similar usage: tail($\Phi(C_1, C_2)$). To find the most stable words (i.e., those whose meaning changed the least), we would instead use head.

### 3.2 Experiments

We compare our method to Hamilton et al. (2016) and use the COHA corpus (Davies, 2010) to contrast word usage in English fiction between $C_1 = 1900$ and $C_2 = 1990$. For word annotations, we used PPMI, SVD, and SGNS (skipgram with negative sampling from Mikolov et al. (2013b)) word vectors released by Hamilton et al. (2016). We set $k = 500$ in the nearest-neighbor scoring function to capture a fair amount of similar words while reducing noise farther down in the neighbor lists.

Table 1 compares the results using the CCLA framework with the semantic change detection described in Hamilton et al. (2016). As with the previous work, we found SVD and SGNS to outperform PPMI. Interestingly, SVD appears to be slightly ahead of SGNS, in contrast to the previous results. Despite this, it has been shown that SVD may be superior to SGNS in some evaluation cases (Levy et al., 2015). Some detected words are shared with those found in Hamilton et al. (*headed*, *gay*) and some words were detected by multiple methods with CCLA (*figured*, *gay*, *handling*, *compound*).

Table 2 shows the nearest-neighbor lists for the words detected to have changed the most by SVD and SGNS. We see that *plane* shifted from meaning a type of inclined or flat surface to a shortened form of *airplane*. The term *figured* changed meaning from describing one's figure (body) to an act of making a decision.

Words that changed the least (i.e., were the most similar) from 1900 to 1990 were non-content words such as *never*, *not*, *eight*, *six*, and *twenty*. These are produced when using head($\Phi(C_1, C_2)$).

## 4 Comparative Lexical Analysis over Context

A context-aware lexical analysis allows us to discover both context-sensitive and context-insensitive terms. Context-sensitive terms are those that may be used to represent their respec-

| Vectors | Method | Top 10 words that changed from 1900s to 1990s |
|---------|--------|-----------------------------------------------|
| PPMI | Hamilton et al. | know, got, would, decided, think, stop, remember, **started**, must, wanted |
| | CCLA | **gay**, favorite, arrangement, please, which, **handling**, random, distributed, available, otherwise |
| SVD | Hamilton et al. | harry, **headed**, **calls**, **gay**, wherever, male, **actually**, special, cover, naturally |
| | CCLA | **handling**, **plane**, **headed**, **gay**, **figured**, **compound**, **kid**, random, **reverse**, division |
| SGNS | Hamilton et al. | **wanting**, **gay**, **check**, **starting**, **major**, **actually**, touching, harry, **headed**, romance |
| | CCLA | **figured**, **guy**, random, **gay**, **chick**, **compound**, **notices**, checking, perspective, **handling** |

Table 1: Comparing methods to find the most-changed words between 1900 and 1990. Each method operates on a type of word representation (PPMI, SVD, or SGNS). We follow the conventions of Hamilton et al. (2016) in bolding terms the authors agree to be clearly correct after consulting a dictionary, underlining borderline cases, and leaving incorrect terms unmarked.

| Word | Vectors | Nearest-neighbors in 1900s | Nearest-neighbors in 1990s |
|------|---------|----------------------------|----------------------------|
| handling | SVD | ribbon, threads, buttons, silk, yellow | delivery, enclosed, send, additional, tax |
| plane | SVD | level, higher, above, horizon, beneath | train, pilot, engines, jet, flight |
| figured | SGNS | thread, lace, rip, lined, stockings | figure, find, thought, pointed, remember |
| guy | SGNS | jane, grey, thomas, chester, roger | tough, person, kid, fellow, man |

Table 2: Example words that changed dramatically during the 20th century. The examples were chosen from the top-20 most-changed lists from words in Table 1.

tive context. For example, *excellent* and *great* could represent a positive sentiment context and *bad* and *horrible* could represent negative sentiment contexts. Context-*in*sensitive terms are those that do not change across contexts, such as stop words. Intelligently assigning scores to these word types will allow us to rank words per context, and even allow us to discover ambiguous words (those whose meaning changes between contexts). Topic models have been used to address some of these issues, and we discuss their differences and limitations in more depth in section 6. Tan et al. (2015) investigated finding ambiguous terms between two corpora, but not in a general contextual text mining framework. In the next sections, we will show how to address these goals with CCLA.

### 4.1 CCLA Formulation

First, we will find ambiguous—or, "context-sensitive"— words between two disjoint contexts in a score-focused manner. We ask the following question: which words' surroundings change the most between $C_1$ and $C_2$? In semantic change detection, $C_1$ and $C_2$ were time periods. Here, we will use contexts from the same time, but with different metadata attributes. Concretely, imagine $D$ is a sentiment analysis dataset. If we let $C_1 = (V_1, f_1)$ where $V_1$ is the set of all words used in positive documents and $f_1(w)$ is a $d$-dimensional word vector learned from only the positive documents, and similarly for $C_2$ with the negative documents, we can discover (1) which words are the most stable between sentiments and (2) which words change the most (i.e., are ambiguous) between sentiments.

We will use the exact same $\phi$ as in section 3:

$$\phi(w, C_1, C_2) = \cos(NN(w, C_1), NN(w, C_2))$$

Now, using $\texttt{head}(\Phi(C_1, C_2))$ we retrieve stable words between sentiment polarities and using $\texttt{tail}$ we discover ambiguous words.

Second, we want to find words that are representative of their context. In the sentiment analysis example, we hope to find words like *amazing* in $C_1$ and *terrible* in $C_2$. To accomplish this, we design a second scoring function $\Phi'$ which uses the previous $\Phi$. We include a third "background" context $C_B$ that covers all the documents in $D$. To find representative words in $C_1$ (i.e., positive words), we use $\texttt{head}(\Phi'(C_1, C_B))$, where

$$\phi'(C_1, C_B) = \phi(C_1, C_B) - \phi(C_1, C_2).$$

The first term compares word contexts in $C_1$ with the background. Recall that $\phi$ gives a high score if the word shares similar neighbors and a low score if the word has different neighbors. A high score may result from two situations: (1) the word's usage is the same in both contexts (e.g., a stop word), or (2) the word's usage is primarily in $C_1$, so when combined with $C_B$, its usage doesn't change.

To filter out the stop words from $\phi(C_1, C_B)$ we subtract $\phi(C_1, C_2)$, since the second term assigns high scores to stable words—stop words. This leaves terms that represent $C_1$ well. Naturally, the same may be done to find words specific to $C_2$.

### 4.2 Experiments

We perform a few different experiments on two popular sentiment analysis datasets, IMDB movie

| | | | |
|---|---|---|---|
| IMDB | Pos | shift, magnificent, lovingly, observed, heartbreaking, determination, marvelous, tightly, globe, superbly | |
| | Neg | travesty, drivel, utter, inane, pile, abysmal, unfunny, idiotic, nonsensical, wretched | |
| | Amb | loggia, krigie, morton, clifford, griffin, chad, epps, deluise, perkins, ana | |
| Yelp | Pos | trails, gem, heavenly, freshest, wonderfully, gifts, tastings, hike, scrumptious, explore | |
| | Neg | zero, tasteless, edible, flavorless, apology, shitty, lousy, irritated, clerk, error | |
| | Amb | behavior, planner, advertising, reaching, silverware, arrogant, avoiding, gratuity, collections, cc | |

Table 3: Using embedding annotation similarity to discover the top positive, negative, and ambiguous terms for IMDB and Yelp. Each corpus treated independently as a separate CCLA problem.

| Word | Positive phrases | Negative phrases |
|---|---|---|
| loggia | **loggia** is wonderful as tony's boss<br>watching o'connor and **loggia** [...] is pure poetry<br>**loggia** i always enjoyed watching in just seeing him yell | **loggia** played his character so lamely<br>**loggia** is about as heroic as a bored businessman.<br>ridiculous attempt at a hispanic accent. (sorry **loggia**.) |
| krige | great cast with alice **krige** and brian krause<br>alice **krige** plays the borg queen again fantastically<br>played by beautiful and talented alice **krige**. | usually excellent alice **krige** is wasted in this one<br>alice **krige** seems to shoulder the film,<br>**krige** gave the only convincing performance |
| morton | love's rebound with socialite anne **morton** (ruth roman)<br>cannavale, rory culkin, joe **morton**, sandra oh, john<br>and **morton** selden (as oberon's grandfather) | thriller from directors rocky **morton** and annabel<br>**morton** is too strong an actress to be relegated<br>is a poor replacement for bob **morton**'s charismatic |
| clifford | is a fledgling playwright named **clifford** anderson<br>old student named **clifford** anderson (christopher reeve)<br>impressed with christopher reeve as **clifford** anderson. | dumps louque for his mate **clifford** grayson<br>her love for his pal, **clifford** greyson (robert noland)<br>his companion **clifford** grayson. what a yawn-fest |
| griffin | and co-stars **griffin** dunn ('after hours')<br>send his younger brother (**griffin** dunne) to law school<br>**griffin** dunne is very well cast as the man | how did lee and **griffin** become such deep friends<br>put together by the hack **griffin** jay who wrote<br>his awful behavior, peter **griffin** has no excuse. |

Table 4: Usage of the five most ambiguous words (all actors and actresses) in the IMDB dataset. In some cases, the same person is discussed in different ways; in others, people share the same name, leading to ambiguity.

reviews and the Yelp academic dataset. For all experiments, we used 300-dimensional word vectors as term annotations that were learned by GloVe (Global Vectors by Pennington et al. (2014)) with the following untuned parameters: window size = 15, max iterations = 25, and a minimum term count of 10 in each corpus.

Table 3 shows two separate CCLA experiments. In each case, we set $C_1$ = terms from positive documents, $C_2$ = terms from negative documents, and $C_B$ = terms from the entire dataset. As in section 3, we set $k = 500$ for the nearest-neighbor lists. We use $\Phi$ to discover ambiguous words and $\Phi'$ to find representative words.

As expected, words such as *travesty* describe negative tones: "a hopelessly miscast, misdirected travesty of actors." At first glance, *shift* may seem a strange choice for positive feelings, but when examined in context, it makes sense: "display her native rhythm and ability to shift tempo in the lavish production" and "a 180-degree shift from the idealistic rhetoric portrayed in [other] offerings."

Ambiguous words offer hints at sentiment targets. In IMDB, the most ambiguous terms are all names of actors and actresses. Table 4 shows example sentences where these words are used. In Yelp, the ambiguous terms are more varied; staff *behavior* could be good or bad and *silverware*

could be clean or dirty. Credit cards ("CC") may or may not be accepted.

Table 5 compares different context views that span both IMDB and Yelp. The "Shared" row sets $C_1$ = all words in positive documents, $C_2$ = all words in negative documents, and $C_B$ = words across all documents in both datasets.

The "Pos only" and "Neg only" rows split $C_i$ by positive and negative documents across both corpora. Ambiguous words in these two rows refer to distinguishing terms between all positive documents based on the corpus or all negative documents based on the corpus.

For example, we can learn the following from this analysis: (1) *magnificent* is used similarly in both datasets for positive sentiment; (2) *unparalleled* is used differently in terms of positive and negative sentiment in both datasets; (3) *disturbing* can be a positive word to describe movies[4], but is not a positive way to describe businesses; (4) *helpful* can be a positive word to describe businesses, but is not a positive way to describe movies; (5) *overzealous* is a negative word in both datasets, but used differently in IMDB vs. Yelp.

When comparing across corpora, we have

---

[4]"This movie is both disturbing and extremely deep" "...very compelling, even disturbing, a chill ran down my spine."

| | | Pos | magnificent, marvelous, breathtaking, heavenly, understated, splendid, exquisite, timeless |
|---|---|---|---|
| Shared | | Neg | unfunny, incoherent, unimaginative, inane, abysmal, horrid, moronic, atrocious, nonsensical, idiotic |
| | | Amb | unparalleled, panoramic, unmatched, daunting, tantalizing, aligned, soft, hardworking, descriptive, serene |
| | IMDB | disturbing, effective, political, engaging, brutal, dramatic, touching, powerful, striking, shocking |
| Pos | Yelp | helpful, whipped, flaky, polite, generous, fluffy, prompt, quaint, trendy, efficient |
| | Amb | orthodox, gargantuan, accented, mirrored, copious, pungent, textured, sweltering, conscientious, kooky |
| | IMDB | corny, contrived, unbelievable, unoriginal, convincing, wealthy, inept, graphic, scary, predictable |
| Neg | Yelp | watery, rubbery, polite, surly, oily, mushy, sticky, helpful, drenched, crusty |
| | Amb | overzealous, ravenous, functional, desolate, impersonal, squashed, callous, grubby, sturdy, blah |

Table 5: Using embedding annotations to compare term contexts between IMDB and Yelp. Cross-corpus lists show words that are used similarly in both collections. Corpus-specific lists show words that are used differently given a particular collection.

the issue of disjoint vocabulary. For example, "movie" is used much more in IMDB reviews than Yelp, even though the term occurs in both. Thus, when comparing positive reviews, "movie" will seem like it's a positive word for IMDB. To combat this, we filter the lists from each cross-corpus analysis, only keeping adjectives.

Since each word is scored with respect to its context, it is a natural extension to use these scored terms in feature selection or even to estimate word polarity scores. Further, scoring terms based on sensitivity to different contexts can be very useful for domain adaptation and transfer learning since we can treat both the source domain(s) and the target domain as contexts to identify terms semantically "stable" across domains, which are intuitively more generalizable than terms very sensitive to domain variations. We would expect shared positive and negative terms between IMDB and Yelp to aid in other sentiment analysis tasks, where the corpus-specific terms are less helpful. The fact that this works even when there is no labeled data in the target domain results in a completely *unsupervised* way to received specialized knowledge.

## 5 Comparing Word Annotations

It is educational so study how annotations drawn from the same data are similar or different. There are many ways to compare embedding methods as annotations using downstream tasks like word analogies or word similarity scoring (Levy et al., 2015). But is there a way to explicitly compare the structure learned by these models? If we have a quantification of this structure, does it give any information about task performance? Levy et al. (2015) consider different word embedding parameters such as adding context vectors (GloVe and SGNS), eigenvalue weighting (SVD), and vector normalization. Other configurations mentioned (but not tested) are number of iterations, vector di-

mensionality, and effect of randomness.

As a demonstration of CCLA's flexibility in choice of context definition, we explore the concept of *word embedding stability*. We define word embedding stability as a measure of how consistent nearest-neighbor lists are across different runs of the same algorithm. Consistency is an important attribute when replicating results or comparing two methods against one another. Different random seeds may play some role in the quality of the word vectors, and methods that use random sampling (like SGNS) may be affected. Nearest-neighbor lists are critical when solving word analogy problems or measuring the similarity between words, so this is the aspect of the word vectors that we will consider while measuring stability.

### 5.1 CCLA Formulation

In sections 3 and 4, we varied the vocabularies for each context. Now, we will vary the word annotations instead in annotation-focused experiments.

Let $C_1$ and $C_2$ represent the same text data ($V_1 = V_2$), but define $f_1(w)$ and $f_2(w)$ as yielding word vectors learned by the same word embedding method with a *different random initialization*. We wish to measure how similar the embeddings are for different runs of the same algorithm.

In the CCLA framework, one way to address this situation requires a similarity metric to measure the nearest-neighbors of the two runs. Before, we used cosine similarity with the term annotation dot product scores as term weights. If we want to stress the orders of the lists themselves, we should ignore the weights and use a ranking correlation metric. The flexibility of CCLA allows us to choose the best measure to suit our task. A rank difference near the top of the lists should be more detrimental than a rank difference farther down the list. In other words, heavy bias should be placed on getting similar top terms to match, rather than terms farther down the list. For this

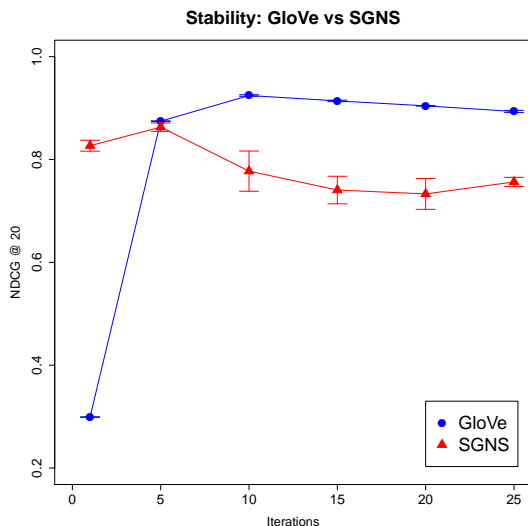

Figure 1: Using NDCG of nearest-neighbor lists to measure the stability of GloVe and SGNS by iteration.

reason, we choose normalized discounted cumulative gain (NDCG) as our measure. Discounted cumulative gain is defined as

$$DCG@n = \sum_{i=1}^{n} \frac{r_i}{\log_2(i+1)}$$

where each element at rank $i$ has a relevance score $r_i$. Normalized DCG divides DCG by the ideal ranking, i.e. sorting the top $n$ elements by their relevance and taking their DCG.

To measure embedding stability, we consider the two ranked nearest-neighbor lists for $w$ from $C_1$ and $C_2$. Without loss of generality, call $C_1$'s list the ideal ranking and assign the relevance scores $n, n-1, \ldots, 1$ to the top $n$ items. We then measure NDCG of $C_2$ with respect to $C_1$'s neighbors as a rank correlation metric, defining the function NDCG@$n(w, C_1, C_2)$. Therefore, stable methods will have a higher average NDCG@$n$ than less stable methods. We can now state $\phi$ for embedding stability measurement as

$$\phi(w, C_1, C_2) = \text{NDCG@}n(w, C_1, C_2)$$

and overall stability score average($\Phi(C_1, C_2)$). Note that we can use this framework to compare embeddings not only from different seeds, but from different algorithms or even dimensions. This measure could be used to see how similarly two or more algorithms perform on the same data.

| Dataset (task) | SGNS | | GloVe | |
|---|---|---|---|---|
| | Low | High | Low | High |
| Google (analogies) | 38.70 | **44.74** | 11.98 | **26.52** |
| MSR (analogies) | 53.41 | **56.32** | 13.92 | **31.92** |
| MEN (word sim) | 51.85 | **58.06** | 23.80 | **38.24** |
| Rare (word sim) | 44.37 | **58.09** | 29.18 | **42.92** |

Table 6: We compare the effect of stability (low vs. high) using analogy and word similarity benchmarks. While stability does not seem to indicate performance differences across embedding methods, it does suggest that a higher stability indicates higher performance within-method.

## 5.2 Experiments

We measure the stability of both GloVe[5] and SGNS[6] at various numbers of iterations and test whether stability may be an indicator of task performance. We used 300-dimensional embeddings trained on the IMDB dataset. In both cases, we used a symmetric window of size 8 with the remaining parameters set to their defaults. For the NDCG measure, we set $n = 20$ to stress performance at the top of the nearest-neighbor lists.

Figure 1 shows the CCLA stability scores from 1 to 25 iterations. Each point on the chart is the average of 10 different random seeds with error bars denoting the standard deviation of the stability scores. SGNS is initially stable, but starts to drop as iterations increase, perhaps indicative of overfitting or model divergence. GloVe's word vectors are fairly consistent after 10 iterations.

We used standard benchmarks for word analogy solving and word similarity scoring. Google analogies (Mikolov et al., 2013a) and MSR analogies (Mikolov et al., 2013c) are written in the form "$a$ is to $b$ as $c$ is to $d$" (where $d$ must be determined). The MEN (Bruni et al., 2012) and Rare (Luong et al., 2013) word similarity tests present word pairs with human-assigned similarity scores. This task is evaluated by measuring the embedding similarity scores' correlation with human judgements via Spearman's $\rho$.

Table 6 compares task performance on embeddings with low stability vs. high stability. For SGNS, we used iteration 25 as the low stability point and iteration 5 as the high stability point; for GloVe, we used 5 as low and 10 as high. SGNS outperformed GloVe in all tasks, even at low stability. Thus, comparing stability across methods may not be a viable metric at suggested performance. Despite this, looking *within-*

---

[5] url redacted for anonymous submission

[6] https://bitbucket.org/yoavgo/word2vecf

method, CCLA's stability measure does seem to indicate that lower-stability runs do underperform the higher-stability runs. This is an especially interesting result for SGNS, since the high stability point is actually at a much lower number of iterations. This suggests that we might use stability as an early-stopping criterion when learning the word representations, potentially saving much compute time while increasing performance.

## 6 Related Work

Our work spans several areas of research:

**Detecting semantic change**. Hamilton et al. (2016) suggest orthogonal Procrustes to align word embedding spaces learned from different time periods, in contrast to per-word heuristics for the alignment (Kulkarni et al., 2015). Kim et al. (2014) start at time period $t$ and learn embeddings. They initialize time period $t + 1$ with those from $t$, and measure which words' cosine similarities changed the most. Unlike the previous two works, this does not produce a mapping function. We propose an approach that does not require embedding matrix alignment and thus does not require an optimization algorithm; we utilize within-period word similarities to create word representations that are comparable across time. This also removes the constraint of incrementally retraining the embeddings each time step; instead of learning 10 embeddings to compare between $t_1$ and $t_{10}$, we learn two and directly compare them with CCLA.

**Contextual text mining**. Topic models have been extended to support analysis of topic variations over different contexts in many ways. In CPLSA (Mei and Zhai, 2006), a generalized form of Zhai et al. (2004), context is incorporated into a topic model as explicit variables. A flexible way to incorporate arbitrary features into a topic model, Dirichlet-multinomial regression, was proposed by Mimno and McCallum (2008). Related recent work is the differential topic model (Chen et al., 2015). There are many topic models for supporting topic analysis in association with specific context such as time and location (e.g., Mei et al. (2006); Yuan et al. (2013)). A common idea in all these and other methods is to model the association of context and topics as word distributions, facilitating cross-context *topic* analysis, but cannot easily support cross-context *lexical analysis*, which is our main goal. An important difference between our work and these contextual topic mod-

els is that our approach does not make parametric assumptions in modeling text (which are generally needed in topic models) and is very flexible, allowing it to easily work with any context and context-specific word annotations.

**Word embedding evaluation**. Word embeddings like SGNS (Mikolov et al., 2013b)) and GloVe (Pennington et al., 2014) have become standard repertoire in text mining and NLP. Some work has been done examining the methods and parameters themselves (Levy and Goldberg, 2014; Levy et al., 2015). Faruqui et al. (2016) find issues with using word similarity as evaluation for embeddings, and suggest only to consider downstream task performance. Our method is able to compare the embedding spaces themselves, which may be a useful alternative to premade similarity datasets or the less direct application tasks.

## 7 Conclusions

We propose a general way to perform cross-context lexical analysis to accommodate *any* notion of context, *any* similarity function, and *any* type of word annotation. This enables many new applications all under the same framework (e.g. development of a common toolkit to support all applications), including analysis of semantic change, comparative analysis of meaning over context, and word embedding stability evaluation.

CCLA opens up interesting new directions for further study, especially in additional applications. One use is to investigate framing bias on political viewpoints. Another is a more fine-grained comparative analysis over specific products as opposed to movies or businesses. Term scoring can be further taken advantage of in sentiment valence prediction. Pablos et al. (2016) use word vector similarity to create sentiment valence scores per term, but they only consider similarity with a manually-chosen positive and negative word. Word sense disambiguation is another unvisited technique, and CCLA's notion of context could help determine which words have multiple senses. Using CCLA as a tool in a larger system is desirable, such as learning to automatically partition a corpus to maximize word differences, or using it for event detection when tones shift from a monitored stream. We want to investigate embedding comparisons further using larger training data and automatically determine an optimal dimensionality or window size given new scoring functions.

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
