# Peer review of "Cross-Context Lexical Analysis"

_ACL 2017 — decision unknown_

[Official Review · Reviewer 1 · rating 4 · confidence 4]
soundness 3 · originality 3 · clarity 4 · impact 3 · substance 4 · appropriateness 5 · meaningful comparison 3 · presentation format Oral Presentation

- Strengths: A nice, solid piece of work that builds on previous studies in a
productive way. Well-written and clear. 

- Weaknesses:

 Very few--possibly avoid some relatively "empty" statements:

191 : For example, if our task is to identify words used similarly across
contexts, our scoring function can be specified to give high scores to terms
whose usage is similar across the contexts.

537 : It is educational to study how annotations drawn from the same data are
similar or different.

- General Discussion:
In the first sections I was not sure that much was being done that was new or
interesting, as the methods seemed very reminiscent of previous methods used
over the past 25 years to measure similarity, albeit with a few new statistical
twists, but conceptually in the same vein. Section 5, however, describes an
interesting and valuable piece of work that will be useful for future studies
on the topic. In retrospect, the background provided in sections 2-4 is useful,
if not necessary, to support the experiments in section 5. 

In short, the work and results described will be useful to others working in
this area, and the paper is worthy of presentation at ACL.

Minor comments:

Word, punctuation missing?
264 : For word annotations, we used PPMI, SVD, and SGNS (skipgram with negative
sampling from Mikolov et al. (2013b)) word vectors released by Hamilton et al.
(2016).

Unclear what "multiple methods" refers to :
278 : some words were detected by multiple methods with CCLA

[Official Review · Reviewer 2 · rating 3 · confidence 3]
soundness 3 · originality 3 · clarity 4 · impact 3 · substance 4 · appropriateness 5 · meaningful comparison 3 · presentation format Poster

This paper propose a general framework for analyzing similarities and
differences in term meaning and representation in different contexts.

- Strengths:
* The framework proposed in this paper is generalizable and can be applied to
different applications, and accommodate difference notation of context,
different similarity functions, different type of word annotations. 
* The paper is well written. Very easy to follow.

- Weaknesses:
* I have concerns in terms of experiment evaluation. The paper uses qualitative
evaluation metrics, which makes it harder to evaluate the effectiveness, or
even the validity of proposed method. For example, table 1 compares the result
with Hamilton et, al using different embedding vector by listing top 10 words
that changed from 1900 to 1990. It's hard to tell, quantitatively, the
performances of CCLA. The same issue also applies to experiment 2 (comparative
lexical analysis over context). The top 10 words may be meaningful, but what
about top 20, 100? what about the words that practitioner actually cares?
Without addressing the evaluation issue, I find it difficult to claim that CCLA
will benefit downstream applications.